

# Detection and characterization of extreme wind speed ramps

Ásta Hannesdóttir and Mark Kelly

DTU Wind Energy Dept., Technical University of Denmark, Roskilde, Denmark

**Correspondence:** astah@dtu.dk

**Abstract.** The present study introduces a new method to characterize ramp-like wind speed fluctuations, including coherent gusts. This method combines two well-known methods: the continuous wavelet transform and the fitting of an analytical form based on the error function. The method provides estimation of ramp amplitude and rise time, and is herein used to statistically characterize ramp-like fluctuations at three different measurements sites. Together with corresponding amplitude of wind direction change, the ramp amplitude and rise time variables are compared to the extreme coherent gust with direction change from the IEC wind turbine safety standard. From the comparison we find that the observed amplitudes of the estimated fluctuations do not exceed the one prescribed in the standard, but the rise time is generally much longer, on average around 200 s. The direction change does however exceed the one prescribed in the standard several times, but for those events the rise time is a minute or more. We also demonstrate a general pattern in the statistical behavior of the characteristic ramp variables, noting their wind speed dependence, or lack thereof, at the different sites.

## 1   Introduction

The IEC wind turbine safety standard prescribes various models of extreme wind conditions that a wind turbine must withstand during its operational lifetime (IEC, 2005). One of those prescribed models is an extreme coherent gust with direction change (ECD), used for ultimate load prediction. The ECD model is presented in Stork et al. (1998), but with a rather limited description; the model is not shown compared to measurements, but it is said to represent extreme gusts and direction changes of wind speed measurements 'quite well.' However, the ECD prescription was found later by Hansen and Larsen (2007) to give reasonable estimates compared with measurements.

With the increasing rotor size of modern wind turbines, resent research has focused on how the gust models in the IEC standard are unrealistically represented by a uniform wave (Bierbooms, 2005; Bos et al., 2014). In these studies, gusts are defined as extreme fluctuations of stationary and homogeneous turbulence. The gusts are simulated with stochastic simulations and constrained in space to have a finite length scale. Using such gust models for wind turbine load simulations generally results in lower loads than when using the uniform gust models of the IEC standard. The reason is due to the limited length scale of the gusts, and that during the simulations some gust might even miss the blades as they sweep by the rotor. The authors of these studies suggest that the uniform gust models of the IEC standard should consequently be replaced by stochastic gust models.



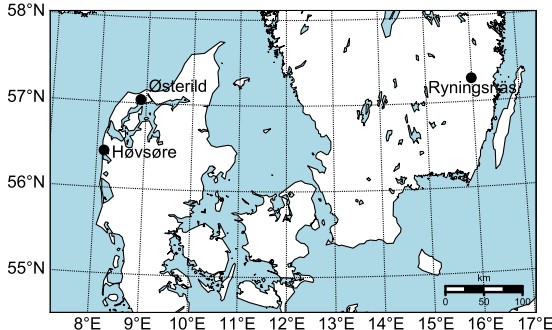

**Figure 1.** A Map of Denmark and southern Sweden showing the locations of the measurement sites.

There are however many studies in the field of atmospheric science that investigate large coherent structures in turbulent flow (e.g. Mahrt, 1991; Belušić and Mahrt, 2012; Barthlott et al., 2007; Fesquet et al., 2009). These studies take into consideration fluctuations of larger scales than those of stationary, homogeneous turbulence, i.e. the submesoscale or mesoscale. These coherent structures are seen in measurements as ramp-like increases in wind speed, that may readily be compared with the ECD due to similar characteristics. The coherent structures can be driven by a broad range of different meteorological processes. In the stable boundary layer they may be generated by e.g. gravity waves, Kelvin-Helmholtz instabilities, surface heterogeneity or pressure disturbances (Mahrt, 2010). In the convective boundary layer they may be generated by e.g. surface buoyancy fluxes, latent heat release or cloud radiative effects and may be observed in form of convective cells and rolls (Drobinski et al., 1998; Young et al., 2002). In the neutral boundary layer they may be generated by shear and can be observed in the form of streaks (Foster et al., 2006). Some processes are bound to certain terrain, e.g coherent structures may be generated by dynamics between the flow and plant canopy (Finnigan, 2000), or in coastal and offshore regions they may be driven by open cellular convection (e.g. Vincent et al., 2012).

In this study we focus on large-scale, high-amplitude (extreme) fluctuations, which are coherent across the rotor of any multi-megawatt wind turbine. We examine data from three sites with different terrain types and characterize the fluctuations. We investigate if the characteristics of the fluctuation are comparable with the ECD. In order to characterize the amplitude and rise-time of the investigated fluctuations we provide a new combination of two well-known methods: the continuous wavelet transform and the fitting of an idealized ramp function (based on the error function), which is inspired by detection of atmospheric boundary-layer depth (Steyn et al., 1999).

## 2 Sites and measurements

The measurements used for the characterization of the ramp-like events come from three different sites. The locations of the measurement sites may be seen in Figure 1.





## 2.1  Høvsøre

The Høvsøre National Test Centre for Wind Turbines is located at the west coast of Jutland, approximately 1.7 km east of the coastline. The site is at a coastal agricultural area where the terrain is nearly flat. Several masts with measurement instruments are located at the site, that has been in operation since 2004. In the current analysis we use measurements from a light mast with cup anemometers and wind vanes installed at 60 m, 100 m and 160 m height. The light mast is located between two of the test wind turbines which are separated by approximately 300 m in the North-South direction. The dominating wind direction is from north-west and the annual average 10-minute wind speed at the light mast is $V_{ave} = 9.33$ m/s at 100 m and the reference turbulence intensity is $I_{ref} = 0.065$ [1]. The data used in this study consists of 10 Hz measurements from September 2004 to December 2014. A detailed overview on the site and instrumentation may be found in Peña et al. (2016).

## 2.2  Østerild

The Østerild National Test Centre for Large Wind Turbines is located in a forested area in Northern Jutland. The distance to the coast is approximately 4 km to the north and 20 km to the west. The site has two 250 m tall light masts equipped with sonic anemometers at 37 m, 103 m, 175 m and 241 m. In this analysis we use measurements from the southern mast, where the terrain around the mast is flat and the surrounding forest has canopy height between 10 and 20 m. To the west of the mast there is a narrow clearing of the forest with a grass field. The clearing is approximately 1 km long in the east-west direction and 200 m wide in the north-south direction. The mast is located approximately 300 m South-west of a row of seven wind turbines aligned in the north-south direction. At the southern light mast, the annual average 10-minute wind speed is $V_{ave} = 7.94$ m/s at 103 m height and the reference turbulence intensity is $I_{ref} = 0.13$. The data used in this study consists of 20 Hz measurements from March 2015 to February 2018 at 37 m, 103 m and 175 m heights. More details on the site may be found in Hansen et al. (2014).

## 2.3  Ryningsnäs

The Ryningsnäs measurement site is located approximately 30 km inland from the south-eastern coast of Sweden. The terrain is forested and generally flat. The forest has a 200 km fetch in the west direction and the tree height around the site is between 20 and 25 m. There is a 138 m tall meteorological mast equipped with sonic anemometers at 40 m, 59 m, 80 m, 98 m, 120 m and 138 m measuring at 20 Hz sampling frequency. In this analysis we use the measurements at 59 m, 98 m and 138 m height from a period between November 2010 and December 2011. There are two wind turbines approximately 200 m from the mast, one in the south direction and the other in the north-east direction. The annual average 10-minute wind speed is $V_{ave} = 5.94$ m/s at 98 m height and the reference turbulence intensity is $I_{ref} = 0.18$. More details on the site and measurements may be found in Arnqvist et al. (2015).

---

[1] $I_{ref}$: The average 10-minute turbulence intensity evaluated at wind speed of 15 m/s



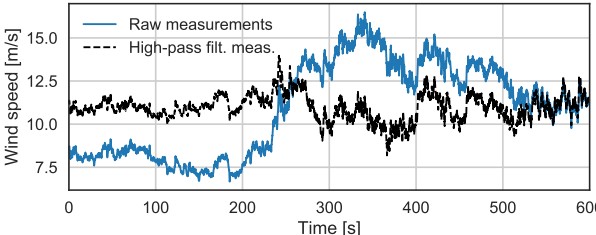

**Figure 2.** Wind speed measurements from 100 m at Høvsøre, raw measurements (blue line) and high-pass filtered measurements (dashed black line).

## 3 Selection and characterization of events

In this section we go through the steps of selecting and characterizing the ramp-like coherent structures. There are three steps in the procedure:

1. Identify events of extreme variance, indicating large scale fluctuations, and acquire 30-minute wind speed measurements for each event.

2. Estimate the time scale and position in time (timing) of the dominating fluctuation using wavelet transform.

3. Characterize the amplitude and rise time of the dominating fluctuation by fitting an idealized ramp function to a subset of the wind speed signal, which timing and scale are found by the wavelet transform.

### 3.1 First step: Selecting high variance events

Here we select the ramp events by comparing two different data sets. One where the 10-minute standard deviation is calculated from the raw measurements, $\sigma_{\mathrm{raw}}$ and the other where the measurements have been high-pass filtered $\sigma_{\mathrm{filt}}$. A significant reduction in the 10-minute standard deviation by high-pass filtering indicates that the measurements include a ramp-like fluctuation. This fluctuation then gives rise to the high observed standard deviation (Hannesdóttir et al., 2018).

The filtering is performed with a second order Butterworth filter where the cut-off frequency is chosen as:

$$f_c = \frac{U}{L} \tag{1}$$

where $U$ is the ten-minute mean wind speed and $L$ is a length scale, here chosen to be 2000 m. With this choice of cut-off frequency the filtered measurements do not include any trends or fluctuations involved with length scales larger than 2000 m.

In order to identify where the 10-minute standard deviation is reduced the most by filtering, we calculate the ratios of $\sigma_{\mathrm{raw}}/(\sigma_{\mathrm{filt}} + 1)$ and identify the highest 0.1% from each data set[2]. We then acquire 30-minute samples of high frequency

---

[2]Here $\sigma_{\mathrm{filt}}$ is shifted by one to put emphasis on high $\sigma$ values. Otherwise only ratios where $\sigma_{\mathrm{filt}} << 1$ are selected.





measurements for each event for further analysis and characterization. By using 30-minute samples we ensure that we have enough measurements before and/or after the ramp-like wind speed increase.

An example of an extreme-variance event may be seen in Figure 2, where 10-minute 'raw' wind speed measurements are compared with filtered measurements. This example is taken from the light mast in Høvsøre at 100 m. The 10-minute standard
deviation of the raw measurements is 2.66 m/s, but 0.75 m/s for the filtered measurements.

### 3.2    Second step: Wavelet transform

The continuous wavelet transform (CWT) unfolds a signal in both frequency and time and provides an efficient way to identify and localize abrupt changes or transients in non-stationary time series. The CWT is often used to identify and characterize coherent structures in turbulent flow (e.g. Dunyak et al., 1998; Krusche and de Oliveira, 2004; Fesquet et al., 2009), or wind
power ramps (Gallego et al., 2013).

The CWT is formally defined as the inner product of a function $x(t)$ and a mother wavelet $\psi(t)$ that is shifted and dilated

$$W_x(\ell,t') = \frac{1}{\ell} \int\limits_{-\infty}^{\infty} x(t)\psi\left(\frac{t-t'}{\ell}\right) dt \qquad (2)$$

where the resulting wavelet coefficients $W_x$ are a function of the scale dilation $\ell$ and time shift $t'$. Note that the factor $1/\ell$ is a normalization resulting in wavelet coefficients in the $L^1$-norm, though this normalization is most commonly seen in the
literature as $1/\sqrt{\ell}$ giving a CWT in the $L^2$-norm (Farge, 1992). However it is important when comparing wavelet coefficients (or wavelet power spectrum) between different scales to do so in the $L^1$-norm, to prevent giving a bias toward the large scales (Liu et al., 2007).

The choice of analyzing wavelet influences the results of the wavelet transform, since it reflects characteristics of the wavelet. We have therefore chosen a wavelet that includes features similar to those we look for in the signal, i.e. one dominating increase
at the center of the wavelet function. The analyzing wavelet chosen here is the first derivative of a Gaussian (DOG1) wavelet[3]

$$\psi(t) = C\,t\,\mathrm{e}^{-t^2} \qquad (3)$$

where $C$ is a normalization constant, here equal to: $2(2/\pi)^{1/4}$. Note that we have switched the sign of the wavelet to get positive wavelet coefficients from the transform where there is an increase in the wind speed signal (Figure 3 (b)).

Figure 3 shows an example of a CWT of one of the detected high-variance events along with the mirrored DOG1 wavelet.
The highest wavelet coefficients are shown with red, indicating a high correlation between the signal and the wavelet at that given time. The maximum wavelet coefficient of the CWT identifies the timing ($t'$) and the scale ($\ell$) of the coherent structure.

### 3.3    Third step: Idealized ramp function

The definition of the idealized ramp function is borrowed from Steyn et al. (1999), where they incorporate the error function into an idealized backscatter profile. The profile is fit to backscatter lidar measurements to identify the depth of the atmospheric

---

[3]The wavelet transform is performed using the Python package PyWavelets (Lee et al., 2006–)





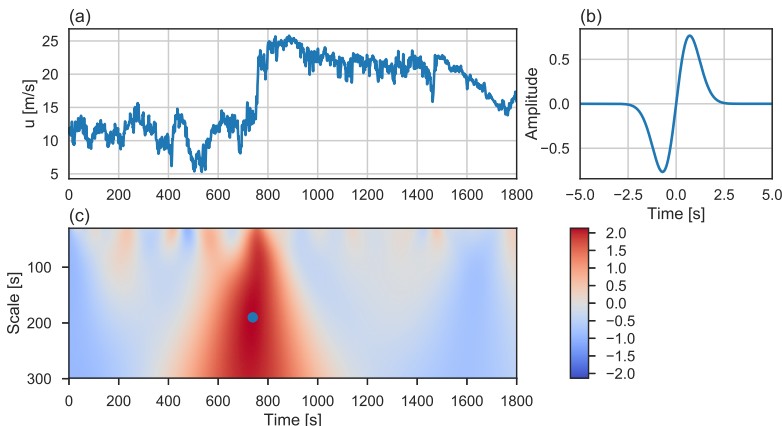

**Figure 3.** The continuous wavelet transform of a ramp-like coherent structure. (a): 30-minute wind speed signal at $100\,\mathrm{m}$, Høvsøre. (b): The flipped DOG1 wavelet used for the wavelet transform. (c): The wavelet coefficients of the wind speed signal. The maximum coefficient is shown with a blue dot at $\ell = 190\,s$ and $t' = 739\,s$

boundary (mixed) layer, and the thickness of the entrainment zone. Wind speed measurements where the wind speed rapidly increases may often resemble these ideal backscatter profiles, and therefore we can use this method to characterize ramp-like fluctuations in the same manner. The idealized ramp wind speed function, may be defined as:

$$u(t) = \frac{u_b + u_a}{2} - \frac{u_b - u_a}{2}\mathrm{erf}\left(\frac{t - t'}{\tau}\right) \tag{4}$$

5 where $u_b$ is the wind speed before the rise, $u_a$ is the wind speed after the rise and $\tau$ is a normalization constant. We define the rise time of the ramp from the interval where the wind speed rises from $0.025u_b$ to $0.975u_a$. This value may be estimated by multiplying $\tau$ with 3.17, which is found from ordinates of the error function. The parameters of the idealized ramp function are found by minimizing the least square differences between the measurements and the ramp function with an optimization curve fitting procedure[4].

10    Figure 4 demonstrates the idealized ramp function that is fit to wind speed measurements from the different sites. The limited period that the ramp function is fit to is found by the CWT. The timing is given by $t'$ and the period is three times the scale: $3 \cdot \ell$. The factor of three is used to ensure approximately equal periods of measurements before, during, and after the ramp-like increase for the curve fitting procedure.

## 3.4 Overview of the selection and characterization

15 A brief summation of the detection: A subset of extreme variance events is found. The CWT is performed on each event and the timing and scale of the ramp-like wind speed increase is estimated. The scale ($3\ell$) is used to find a limited period of the

---

[4]For the optimization fitting procedure we employed the SciPy curve_fit function





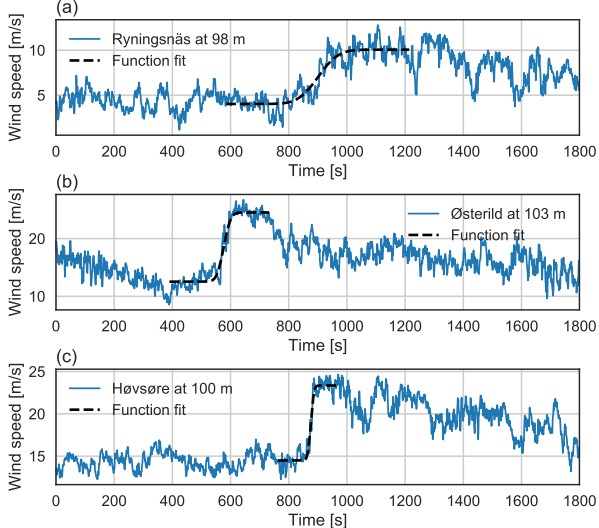

**Figure 4.** Three examples of the idealized ramp function fit to the wind speed measurements. Measurements from: (a) Ryningsnäs (b) Østerild (c) Høvsøre. The blue lines shows the measurements and the dashed black lines shows the idealized ramp function fit to a subset of the measurement.

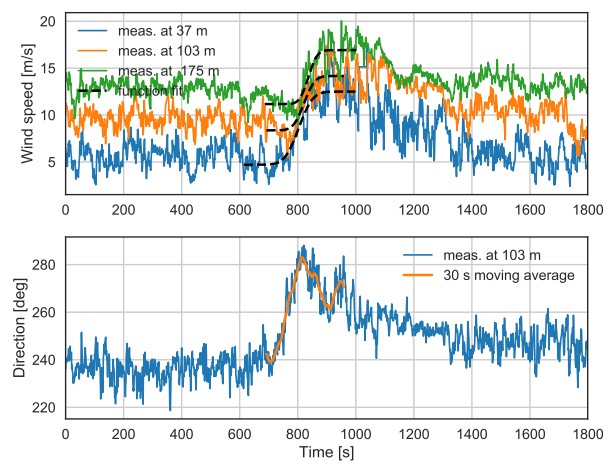

**Figure 5.** The idealized ramp function fit to Østerild measurements at three different heights and the corresponding direction change.

wind speed signal to which the idealized ramp function is fit. The idealized ramp function parameters are used to estimate the amplitude of the ramp-like fluctuation: $\Delta u = u_a - u_b$, and the rise time: $\Delta t = 3.17\tau$.





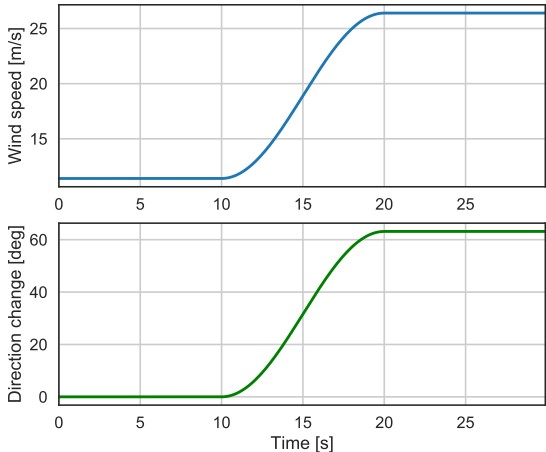

**Figure 6.** The extreme coherent gust with direction change from the IEC wind turbine safety standard.

As we want to compare the wind speed ramps with the ECD load case of the IEC standard, we investigate the direction change during the ramps. Here we use the directional data at $\approx 100\,\text{m}$ from each site and calculate the moving $30\,\text{s}$ average during the time of the ramp function at $\approx 100\,\text{m}$.

The direction change during the ramp-like wind speed increase is determined as the difference between the maximum value and the minimum value of the moving average.

An example of an ramp event at Østerild is shown in Figure 5 along with the corresponding directional data. The orange line in the lower panel shows the $30\,\text{s}$ moving average during the ramp function period at $103\,\text{m}$. The moving average is applied to the directional measurements in order to filter out the small scale fluctuations that we do not want to influence the estimated direction change.

The amplitudes and rise times are characterized for each measurement height. Afterwards the values are averaged over the three different heights to give the characteristic rise time and amplitude for each event.

## 4 IEC extreme coherent gust with direction change

The extreme coherent gust with direction change (ECD) is modeled with an amplitude of $V_{cg} = 15\,\text{m/s}$ and a direction change

$$
\theta_{cg} = \begin{cases} 180°, & \text{if } V_{\text{hub}} \leq 4 \text{ m/s.} \\ 720° \,[m/s]/V_{\text{hub}}, & \text{if } 4\,\text{m/s} < V_{\text{hub}} < V_{\text{ref}}, \end{cases} \tag{5}
$$



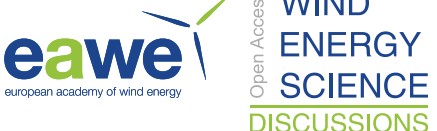

where $V_{\mathrm{hub}}$ is the 10-minute mean wind speed at hub height and $V_{\mathrm{ref}}$ is the 10-minute mean reference wind speed. Both the direction change and wind speed change are modeled as functions of time,

$$\theta_{\mathrm{cg}}(t) = \begin{cases} 0^\circ, & \text{if } t < 0 \\ \pm 0.5\theta_{\mathrm{cg}}(1 - \cos(\pi t/T)), & \text{if } 0 \leq t \leq T \\ \pm\theta_{\mathrm{cg}}, & \text{if } t > T \end{cases} \tag{6}$$

$$V(z,t) = \begin{cases} V(z), & \text{if } t < 0 \\ V(z) + 0.5V_{\mathrm{cg}}(1 - \cos(\pi t/T)), & \text{if } 0 \leq t \leq T \\ V(z) + V_{\mathrm{cg}}, & \text{if } t > T \end{cases} \tag{7}$$

where $T = 10\,s$ is the rise time. The direction change and wind speed increase are assumed to occur simultaneously. Figure 6 shows the ECD for $V_{\mathrm{hub}} = V_r = 11.4\,m/s$ which is the rated wind speed for e.g. the NREL 5 MW- and the DTU 10 MW reference wind turbines (Jonkman et al., 2009; Bak et al., 2013). According to the IEC standard, the design load case with the ECD should be simulated at $V_r \pm 2$ m/s.

## 5   Distributions and comparison with the ECD

In this section we look at the amplitudes, rise times and direction change of the detected events and how these variables are distributed. Selecting the 0.1% highest ratios of $\sigma_{\mathrm{raw}}/(\sigma_{\mathrm{filt}} + 1)$ results in 453 events from Høvsøre, 154 from Østerild and 58 from Ryningsnäs. A number of these events are discarded before performing the characterization, for one of three reasons: because the measurements are partly missing; because the measurements are from a wind direction sector where the nearby

wind turbines are upstream of the masts (in the wake of the wind turbines); or because the high observed variance is due to a wind speed decrease (negative ramps). The negative ramps are identified when the dominating wavelet coefficients are negative. The discarding narrows the number of analyzed events down to 216 from Høvsøre, 72 from Østerild and 32 from Ryningsnäs.

The estimated $\Delta u$, $\Delta\theta$ and $\Delta t$ variables for each detected event and their distribution may be seen in Figure 7. The variables

are shown with different colors for each measurement site, black for Høvsøre, blue for Østerild and green for Ryningsnäs. It may be seen that the highest values of each parameter are found from the Høvsøre data set, that has the longest measurement period.

The sample-means and the corresponding standard deviations of $\Delta u$, $\Delta\theta$ and $\Delta t$ for each site may be found in Table 1. Though the variables are not normally distributed, we choose to show the standard deviation to indicate the spread of the

variables. The average $\Delta u$ and $\sigma_{\Delta u}$ are of similar magnitude for all sites. We see that the average $(\Delta\theta)$ and standard deviation of direction change $(\sigma_{\Delta\theta})$ found in Ryningsnäs is nearly twice the value found at Østerild and significantly higher than at Høvsøre.





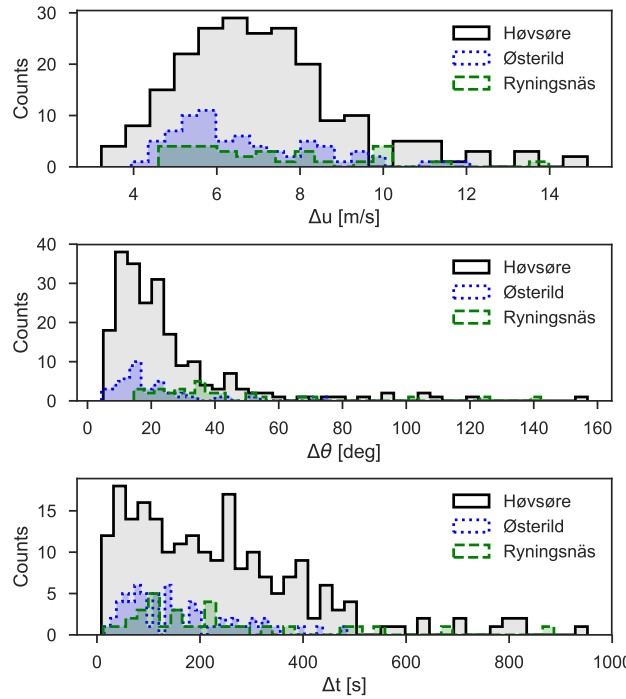

**Figure 7.** The distributions of detected amplitudes ($\Delta u$), direction changes ($\Delta \theta$) and rise times ($\Delta t$) of all the detected events at the three different sites: Høvsøre (grey), Østerild (blue) and Ryningsnäs (green).

|  | Høvsøre | Østerild | Ryningsnäs |
|---|---|---|---|
| Nr. of analyzed events | 216 | 72 | 32 |
| $\langle \Delta u \rangle \pm \sigma_{\Delta u}$ | 7.1± 2.1 m/s | 6.7± 1.8 m/s | 7.3± 2.2 m/s |
| $\langle \Delta \theta \rangle \pm \sigma_{\Delta \theta}$ | 25± 21° | 21± 14° | 42± 30° |
| $\langle \Delta t \rangle \pm \sigma_{\Delta t}$ | 232± 177 s | 160± 105 s | 233± 195 s |

**Table 1.** The number of analyzed events and average estimated variables from each site.

The average $\Delta t$ and $\sigma_{\Delta t}$ are lowest for the Østerild site, and there are no events detected with rise time above 485 s, while the maximum estimated rise time in Ryningsnäs and Høvsøre are 887 s and 952 s respectively.

Figure 8 shows the detected events as function of mean wind speed compared with the ECD model. The mean wind speed is the average of $u_b$ and $u_a$, which may be taken as the representative wind speed of the events. A similar figure has been made showing the events as function of $u_b$, and may be found in Appendix A. The dashed lines show the IEC prescription of $\Delta u$, $\Delta \theta$ and $\Delta t$ for the ECD. The solid lines show the variables averaged over wind speed bins where the bin width is 2 m/s. The shaded colors mark the area between the 10th percentile and the 90th percentile of the variables in each bin. When comparing the estimated $\Delta u$ to the IEC prescribed amplitude, it is seen that there is not a single event that exceeds 15 m/s. There is a




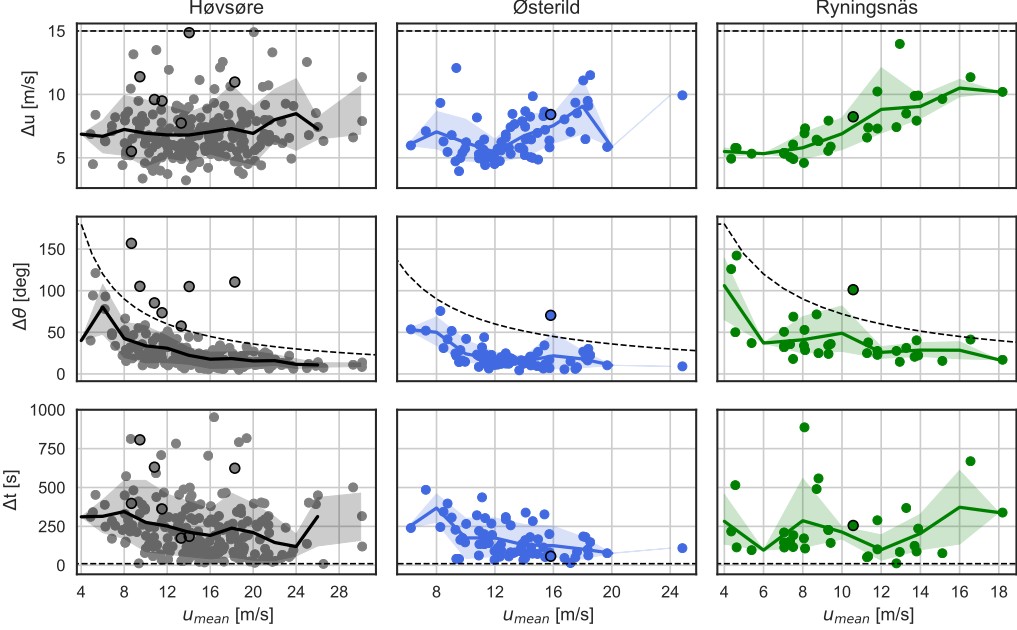

**Figure 8.** The detected amplitudes ($\Delta u$), direction changes ($\Delta\theta$) and rise times ($\Delta t$) as function of the mean wind speed at the different sites: Høvsøre (grey), Østerild (blue) and Ryningsnäs (green). The plots share the primary axis (abcissa) per column and they share the secondary axis (ordinate) per row. Shaded area indicates events between 10th percentile and 90th percentile for each wind speed bin, with bin width 2 m/s. The black dashed lines show the ECD values.

number of events that exceed the prescribed direction change of the ECD, one at Ryningsnäs, one at Østerild and seven at Høvsøre. These extreme direction events are indicated with a black circle in the different plots. The amplitude of the extreme direction events in Høvsøre ranges from $\Delta u = [5.5, 14, 9]$ m/s and the rise times range from $\Delta t = [174, 807]$ s. The extreme direction event at Østerild has a direction change of $\Delta\theta = 70°$, an amplitude of $\Delta u = 8.4$ m/s and a rise time of $\Delta t = 58$ s. At

5    Ryningsnäs the extreme direction event has $\Delta\theta = 101°$, $\Delta u = 8.2$ m/s and $\Delta t = 256$ s.

## 6    Discussion

### 6.1    Discussion on the detection and characterization method

The CWT is ideal to find abrupt changes in a wind speed signal and can provide useful information on different scales of the flow. Here we use the wavelet transform to provide an objective estimate of the time scale of the ramp-like wind speed increase

10    as well as the precise timing in the signal. To obtain characteristics of the amplitude and rise-time of these fluctuations we need an additional step, which is inspired by mixed layer height detection performed by fitting an idealized profile to backscatter measurements.





The main difference between backscatter profiles and wind speed time series, is that the wind speed continuously fluctuates through time and the period of the coherent structure we investigate is finite. This difference is why the wavelet analysis is important prior to the fitting of the idealized function, where the limited period of the ramp and the timing is identified. We found the optimal period for the fitting to be three times the scale dilation ($3 \cdot \ell$) of the DOG1 wavelet as defined in section 3.2.

If this limited period is not long enough, the numerical curve fitting procedure might not always find an optimal solution to the fitting parameters. Having enough measurement points for a curve fitting procedure is what makes the method robust as pointed out by Steyn et al. (1999). However, for the purpose of characterizing wind speed fluctuations it is important that the chosen fitting period is not longer than necessary. We see e.g. for the wind speed fluctuations in Figure 4, that the wind speed decreases shortly after the ramp; if this decrease is included in the curve fitting, the amplitude of the estimated ramp would

be underestimated. The choice of $3 \cdot \ell$, provides the shortest period that makes the combined method robust in the sense that it always results in a successful fit with an estimate of the desired parameters.

The first step in the selection, choosing high variance events, is used for two purposes: First, to ensure that the selected ramp-like fluctuations are associated with scales that are large enough to cover any rotor of a multi-megawatt wind turbine. We have seen in a previous study that these fluctuations occur approximately simultaneously at two different measurement

masts in Høvsøre that are separated by 400 m transverse to the mean wind direction (Hannesdóttir et al., 2018). Second, by choosing a subset of events, we avoid performing a CWT on the whole data set of high frequency measurements, which is computationally demanding on a 10-year data set like the one from Høvsøre. If a CWT is performed on the whole data set an extra step would be needed in the analysis to decide whether a structure is coherent or not, e.g. to apply a threshold on the scale averaged wavelet coefficients or wavelet spectrum (e.g. Farge, 1992; Dunyak et al., 1998).

**6.2   Discussion of observed distributions**

The main difference between the observed fluctuations analyzed in the current study and the classic ECD (investigated in Stork et al., 1998; Hansen and Larsen, 2007), is that in the current study we only characterize large scale coherent structures, whereas the ECD is based on measurements where all extreme peaks of small-scale turbulence are considered. By extracting ramp events from the measurements, we exclude the small-scale fluctuations from the characterization of the amplitude and rise

time (see Figures 4 and 5). Even though Hansen and Larsen (2007) only consider 10 s rise times from a data set with a two year period, they find gust amplitudes in a similar range as in the current study. This is because small-scale turbulent fluctuations can have very high peak values. However, such fluctuations are not coherent across rotor diameters of multi-megawatt wind turbines, and have much less impact than coherent ramps on loads for such turbines.

We observe that the average amplitudes of ramp-like fluctuations ($\langle \Delta u \rangle$) is of similar magnitude at all the sites considered.

As shown in Fig. 8, $\Delta u$ has negligible wind speed dependence at Høvsøre and Østerild, but at Ryningsnäs the ramp amplitudes increase with mean wind speed. The direction change generally decreases with wind speed at all the sites, but significantly larger mean change $\langle \Delta \theta \rangle$ is observed over ramps at Ryningsnäs. These observations are consistent with the (low-order, dominant) physics of the sites: Ryningsnäs has appreciably taller trees than Østerild, with the Ryningsnäs observations taken at roughly 2–5 times tree height; the measurements used from Østerild correspond to 5-15 times the respective tree heights there.




Thus the measurements at Ryningsnäs are more affected by the tree-induced turbulent stresses (e.g. Raupach et al., 1996; Sogachev and Kelly, 2016). In particular, a wind-speed (Reynolds-number) dependence arises in the turbulent degradation of the coherent structures, and there is more turning of the wind due to the relatively larger drag.

**Are the ramps comparable to the ECD?**

The rise time of the ramp-like fluctuations is generally much higher than that of the ECD. But the range is large, e.g. at Høvsøre the rise time ranges over two orders of magnitude (from 9–952 s). The rise time of the extreme direction events is on the order of a minute or more. Although these extreme direction events generally have a longer rise time than the defined ECD, they could readily be considered for load simulation purposes. The reason is that a wind turbine reacts much slower to changes in wind direction than to changes in wind speed. The yaw speed of a wind turbine is typically less than $0.5°/s$, which means

that yawing $90°$ takes more than 3 minutes. Hence, during one of the extreme direction events, a wind turbine is continuously exposed to yaw misalignment, while the wind speed keeps increasing.

We observe ramp events that either have an amplitude, or rise time, or direction change on the same order of magnitude as the ECD. However, no single event is comparable to the ECD on all three variables at once. In order to predict an extreme event considering all three variables simultaneously, one would need a multivariate distribution model including the parameter

distributions. That way it would be possible to model the probability of different positions in the three-parameter space and extrapolate to desired return periods.

## 7   Conclusions

The combination of the wavelet transform and the fitting of an idealized ramp function is a new and efficient way to characterize extreme wind speed ramps. The characterization provides variables that are relevant for wind energy, particularly for wind

turbine load simulations, probabilistic design, and wind turbine safety standards.

We use measurements from three measurement sites in different terrain to calculate statistics of the amplitudes, direction change and rise time of extreme ramp-like fluctuations, and also compare the estimated variables with the ECD load case of the IEC standard. Here we find:

- The amplitudes of these coherent structures do not exceed the amplitude of the ECD (using ten, three, and one year of
data, respectively).

- The amplitudes show no clear wind speed dependence at Høvsøre and Østerild, but at Ryningsnäs the amplitudes increase with increasing wind speed.

- The direction change may exceed that of the ECD, but for those events the rise time is a minute or more.

Future related work includes further analysis of ramp events. In particular, using a multivariate distribution model based on

the marginal distributions of the ramp variables to estimate ramp events with a 50-year return period.



*Data availability.* The data is in an SQL database at DTU that is not publicly accessible.

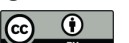



## Appendix A

The Figure in this appendix is equivalent to Figure 8, but shows the estimated variables as function of the speed $u_b$ preceding the ramp.

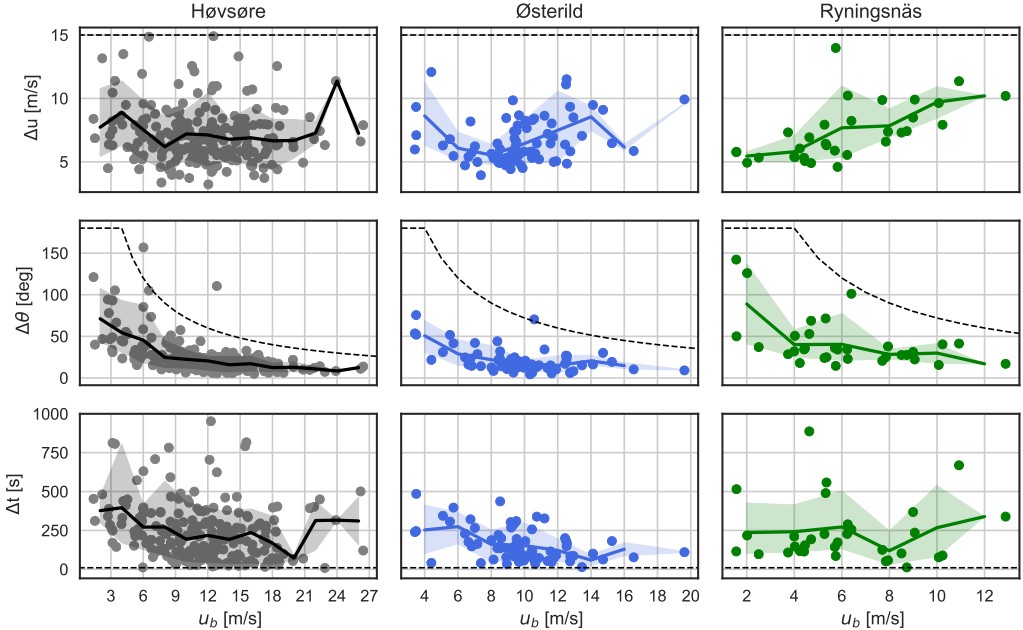

**Figure A1.** The detected amplitudes ($\Delta u$), direction changes ($\Delta \theta$) and rise times ($\Delta t$) as function of speed preceding the ramp ($u_b$) at the different sites: Høvsøre (grey), Østerild (blue) and Ryningsnäs (green). The plots share the primary axis (abcissa) per column and they share the secondary axis (ordinate) per row. The black dashed lines show the ECD values.

Note the IEC direction change prescription looks more reasonable when using $u_b$. This is because $u_b$ is lower than the the average of $u_b$ and $u_a$ and the events get shifted to the left by using $u_b$ when compared with Figure 8. This difference is greatest for the large amplitude events.

*Author contributions.* ÁH provided the detection method, performed the data analysis, and made the figures. MK provided guidance and comments. ÁH prepared the manuscript with contributions from MK.

*Competing interests.* The authors declares that they have no conflict of interest.



*Acknowledgements.* This work is part of ÁH's PhD under supervision of MK funded by DTU Wind Energy.



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
