# Peer review of "Detection and characterization of extreme wind speed ramps"

_Wind Energy Science, 2018_

## Referee Comment (RC1) · Anders Wickström (Referee) · 4 Feb 2019

General comments The design load case ECD is part of the international design standard for wind turbines, IEC61400-1, which makes the topic of the paper of broad international interest. It is very good that the deterministic wind conditions are analysed and compared to real measurements. Especially the ECD seems to be rather arbitrarily defined in the early days. The conditions at ECD are normally not a design driver load case. Maybe that is a reason the parameters and conditions for EDC have not attended the most interest amongst wind turbine designers or manufacturers. Finding methods to validate the ECD seems also difficult, which is discussed in the paper. One method is proposed and presented, which is a good attempt to get clarification in the subject. The focus is on large-scale, high-amplitude fluctuations, which are coherent

across the rotor of multi-megawatt wind turbines, which is relevant as the turbine sizes has increased significantly since the ECD was first proposed for wind turbine design.

Specific comments The measurements used for the characterization of the ramp-like events come from three different sites, located in a quite narrow geographical area of southern Scandinavia. What is the motivation for selecting these sites? Are there other sites with potential more complex terrain or other conditions that might lead to more severe gust events? What global wind measurement data is available for this kind of research? When the methods are specified and algorithms are coded, it seems relatively straight forward to compile a larger amount of data, to get even more reliable base for conclusions. For further discussion: It would be interesting to check if there are correlations with ECD severity and the turbulence intensity at the specific site?

Technical corrections Figure 2 should maybe be moved further down in the article, when the high pass filter has been introduced, e.g. after line 13 or before just 3.2. Page 6, line 6 "We define the rise time of the ramp from the interval where the wind speed rises from 0.025ub to 0.975ua." 0.025ub seems very low!? Should it be 0.025ua? No Nomenclature.

---

## Referee Comment (RC2) · Matti Koivisto (Referee) · 3 Mar 2019

In the paper "Detection and characterization of extreme wind speed ramps" the authors present a method for characterizing wind speed ramps. The method uses 3 steps: comparison of filtered and unfiltered 10 min standard deviations (SDs) to find interesting 30 min event windows, wavelet transform to find the interesting events in these windows and a ramp function fit to estimate and characterize the actual ramping events. The method is applied to multiple years of data from three locations. The method is interesting and generally well presented; however, please consider the following comments.

Introduction:

[Figure]

1) It would be good to specify even clearer if the aim of the presented methodology is to: a) Forecast ramp events (the time when they are expected to start and how long they are expected to last), or b) Summarize/specify ramp behavior of measured (historical) data

Figure 2:

2) It would be clearer to move this figure after Section 3.1; I would prefer that all figures are mentioned in the text before they appear in the paper.

Intro/Section 3:

3) A general question: Why is a (relatively) complex 3-step ramp event characterization method needed? Could one just calculate differences from the data, i.e., study variables such as $y\_delta = y\_t - y\_t\text{-lag}$? Considering different lags (e.g., 1 s, 1 min, 10 min, 1 hour), one could study the probability distributions of such $y\_delta$ variables (PDFs, SDs, percentiles, etc.) to characterize wind speed variability (ramping). What does the proposed methodology offer compared to such very simple calculations?

Intro/Section 3.1:

4) Are there any other comparable "ramp event identification" methods in the literature? How does the proposed method, and especially the comparison of filtered and unfiltered 10 min SDs compare to them?

Section 3.1:

5) Are the 10 min SDs calculated using a sliding 10 min window, or using pre-defined 10 min windows (e.g., 00:00-00:10, 00:10-00:20, and so on)?

6) Some of the selections, e.g., taking 10 min means, using L = 2000m and taking 0.1% of the highest values seem a bit arbitrary. Has some validation been carried out that the selection of high variance events really chooses the important ramp events? Maybe comparison to possible other "ramp event identification" methods?

Section 3.1/References:

7) Reference "Hannesdóttir et al., 2018": When clicking the DOI link, I get to paper "Extreme fluctuations of wind speed for a coastal/offshore climate: statistics and impact on wind turbine loads"; whereas in the Reference list it reads: "Extreme wind fluctuations: joint statistics, extreme turbulence, and impact on wind turbine loads". Is this the correct reference? (the title of the reference should be fixed anyway)

Figure 4:

8) What does the proposed method do if there are two interesting ramp events in the same 1800 s window?

Figure 7:

9) Maybe it would be better to show frequencies instead of counts in Figure 7? As now the counts don't seem very comparable due to different measurement periods.

Section 3.4/Discussion:

10) Are the most interesting ramp events such where both wind speed and direction change quickly? It seems that now only wind speed is used when finding the interesting ramp events (and wind direction change is then simply calculated for these already selected events). But could a combination of wind speed and direction be considered in the ramp event detection method to find the most interesting events considering both wind speed and direction change?

---

## Author Comment (AC1) · 10 Mar 2019

**Responses to reviewer Anders Wickström**

We want to start by thanking you for the positive general comments and good specific suggestions that we can use to improve the paper. We will address your comments in a chronological order, where the reviewer comments are with blue italic font.

*General comments The design load case ECD is part of the international design standard for wind turbines, IEC61400-1, which makes the topic of the paper of broad international interest. It is very good that the deterministic wind conditions are*

*analysed and compared to real measurements. Especially the ECD seems to be rather arbitrarily defined in the early days. The conditions at ECD are normally not a design driver load case. Maybe that is a reason the parameters and conditions for EDC have not attended the most interest amongst wind turbine designers or manufacturers. Finding methods to validate the ECD seems also difficult, which is discussed in the paper. One method is proposed and presented, which is a good attempt to get clarification in the subject. The focus is on large-scale, high-amplitude fluctuations, which are coherent across the rotor of multi-megawatt wind turbines, which is relevant as the turbine sizes has increased significantly since the ECD was first proposed for wind turbine design.*

*The measurements used for the characterization of the ramp-like events come from three different sites, located in a quite narrow geographical area of southern Scandinavia. What is the motivation for selecting these sites? What global wind measurement data is available for this kind of research?*

There was not a specific motivation for choosing this geographical area. The reason is rather that at our university we have access to data from these sites and unfortunately not many others. Data is most often privately owned and not freely available.

*When the methods are specified and algorithms are coded, it seems relatively straight forward to compile a larger amount of data, to get even more reliable base for conclusions.*

Yes, we agree and would very much like to have access to more data sets with long term high-frequency measurements at different geographical areas.

*For further discussion: It would be interesting to check if there are correlations with ECD severity and the turbulence intensity at the specific site?*

Yes, this is an interesting point. We do actually get some indication that this is not the case. From the description of the different sites the reference turbulence intensity at Høvsøre is 0.065 and at Ryningsnäs it is 0.18, yet the average ramp amplitudes are of similar magnitude. This is likely because these large coherent structures are caused by mesoscale phenomena, observed at heights above the surface layer.

*Technical corrections Figure 2 should maybe be moved further down in the article, when the high pass filter has been introduced, e.g. after line 13 or before just 3.2.*

Yes, good point. We will change the placement of the figures as far as the Copernicus latex template allows.

*Page6, line 6 "We define the rise time of the ramp from the interval where the wind speed rises from 0.025ub to 0.975ua." 0.025ub seems very low!? Should it be 0.025ua?*

Thank you for pointing this out. This is an error that will be corrected. It should actually be from 0.0125 to 0.9875 of the total amplitude $(u_a - u_b)$.

*No Nomenclature.*

Do you suggest a list with acronyms and/or symbols? This can be included in a reviewed version.

---

## Author Comment (AC2) · 10 Mar 2019

**Responses to reviewer Matti Koivisto**

We would like to thank you for the constructive comments and suggestions. We will address your comments in order, where the reviewer comments are with blue italic font.

*In the paper "Detection and characterization of extreme wind speed ramps" the authors present a method for characterizing wind speed ramps. The method uses 3 steps: comparison of filtered and unfiltered 10 min standard deviations (SDs) to find interesting 30 min event windows, wavelet transform to find the interesting events in*

*these windows and a ramp function fit to estimate and characterize the actual ramping events. The method is applied to multiple years of data from three locations. The method is interesting and generally well presented; however, please consider the following comments.*

*Introduction:*
*1) It would be good to specify even clearer if the aim of the presented methodology is to: a) Forecast ramp events (the time when they are expected to start and how long they are expected to last), or b) Summarize/specify ramp behavior of measured (historical) data*

Good suggestion. We will specify further in the introduction that this method is used to characterize wind speed ramps in wind speed measurements. The focus in the present study is not related to ramp forecasting or wind power ramps in connection with the electric grid, but rather extreme wind speed fluctuations that may be considered for load purposes.

*Figure 2: 2) It would be clearer to move this figure after Section 3.1; I would prefer that all figures are mentioned in the text before they appear in the paper.*

Yes, good point. We will change the placement of the figures as far as the Copernicus latex template allows.

*Intro/Section 3:*
*3) A general question: Why is a (relatively) complex 3-step ramp event characterization method needed? Could one just calculate differences from the data, i.e., study variables such as y_delta = y_t - y_t-lag? Considering different lags (e.g., 1 s, 1 min, 10 min, 1 hour), one could study the probability distributions of such y_delta variables (PDFs, SDs, percentiles, etc.) to characterize wind speed variability (ramping). What*

*does the proposed methodology offer compared to such very simple calculations?*

This is an interesting question and a valid consideration. We have tried similar methods earlier. The main disadvantage of such an approach is that you would have to choose different pre-defined time lags and the method would not provide you with a single characteristic rise time (or time lag) of the ramp. The method would always provide an amplitude (y_delta) estimate for all the pre-defined time lags and how would you know what is the appropriate time lag? One might think the appropriate time lag is the one giving the largest amplitude, but small scale fluctuations will often have a significant impact on such amplitude estimates. The main advantage of the current method is that it gives a characteristic estimate of both the time-scale and the amplitude of the studied ramps. The method finds unique events relevant for loads and is not influenced by, or used to characterize small-scale turbulence.

*Intro/Section 3.1:*
*4) Are there any other comparable "ramp event identification" methods in the litera-ture? How does the proposed method, and especially the comparison of filtered and unfiltered 10 min SDs compare to them?*

Yes, some have been referenced and one will be added. The un/filtered standard deviations are different than the 'fast' ramp characteristics, though these statistics are affected by the ramps and turbulence.

*Section 3.1:*
*5) Are the 10 min SDs calculated using a sliding 10 min window, or using pre-defined 10 min windows (e.g., 00:00-00:10, 00:10-00:20, and so on)?*

Thank you, good question. The 10-minute standard deviations are calculated from pre-defined windows. This information will be added to the paper.

*6) Some of the selections, e.g., taking 10 min means, using L = 2000m and taking 0.1% of the highest values seem a bit arbitrary. Has some validation been carried out that the selection of high variance events really chooses the important ramp events? Maybe comparison to possible other "ramp event identification" methods?*

Yes some choices seem to be arbitrary and even though we generally try to avoid such choices some are unavoidable. Taking 10-minute averaging times is often argued as a good choice within wind energy, as that time scale is in the 'spectral gap'; it is standard practice in our field. Slower (>10 minute) ramps may affect production, but do not appreciably affect loads. The scale L=2000m is at the lower end of the mesoscale range, and in the current study we want to exclude the largest mesoscale fronts from the characterization. Taking 0.1% is an arbitrary threshold, but it only influences the number of detected events.

---

## Author Comment (AC3) · 9 Apr 2019

**Responses to reviewer Matti Koivisto**

We would like to thank you for the constructive comments and suggestions. We will address your comments in order, where the reviewer comments are with blue italic font.

*In the paper "Detection and characterization of extreme wind speed ramps" the authors present a method for characterizing wind speed ramps. The method uses 3 steps: comparison of filtered and unfiltered 10 min standard deviations (SDs) to find interesting 30 min event windows, wavelet transform to find the interesting events in*

*these windows and a ramp function fit to estimate and characterize the actual ramping
events. The method is applied to multiple years of data from three locations. The
method is interesting and generally well presented; however, please consider the
following comments.*

*Introduction:*
*1) It would be good to specify even clearer if the aim of the presented methodology
is to: a) Forecast ramp events (the time when they are expected to start and how
long they are expected to last), or b) Summarize/specify ramp behavior of measured
(historical) data*

Good suggestion. We will specify further in the introduction that this method is used to
characterize wind speed ramps in wind speed measurements. The focus in the present
study is not related to ramp forecasting or wind power ramps in connection with the
electric grid, but rather extreme wind speed fluctuations that may be considered for
load purposes.

*Figure 2: 2) It would be clearer to move this figure after Section 3.1; I would prefer that
all figures are mentioned in the text before they appear in the paper.*

Yes, good point. We will change the placement of the figures as far as the Copernicus
latex template allows.

*Intro/Section 3:*
*3) A general question: Why is a (relatively) complex 3-step ramp event characteriza-
tion method needed? Could one just calculate differences from the data, i.e., study
variables such as y_delta = y_t - y_t-lag? Considering different lags (e.g., 1 s, 1 min,
10 min, 1 hour), one could study the probability distributions of such y_delta variables
(PDFs, SDs, percentiles, etc.) to characterize wind speed variability (ramping). What*

*does the proposed methodology offer compared to such very simple calculations?*

This is an interesting question and a valid consideration. We have tried similar methods earlier. The main disadvantage of such an approach is that you would have to choose different pre-defined time lags and the method would not provide you with a single characteristic rise time (or time lag) of the ramp. The method would always provide an amplitude (y_delta) estimate for all the pre-defined time lags and how would you know what is the appropriate time lag? One might think the appropriate time lag is the one giving the largest amplitude, but small scale fluctuations will often have a significant impact on such amplitude estimates. The main advantage of the current method is that it gives a characteristic estimate of both the time-scale and the amplitude of the studied ramps. The method finds unique events relevant for loads and is not influenced by, or used to characterize small-scale turbulence.

*Intro/Section 3.1:*
*4) Are there any other comparable "ramp event identification" methods in the literature? How does the proposed method, and especially the comparison of filtered and unfiltered 10 min SDs compare to them?*

Yes, some have been referenced and one will be added (Sevlian and Rajagopal 2013). The un/filtered standard deviations are different than the 'fast' ramp characteristics, though these statistics are affected by the ramps and turbulence. We do however not know of any studies that considers ramps of this time scale (within 10 minutes) that we can compare with. The referenced ramp detection studies (Gallego et al 2013 and Sevlian and Rajagopal 2013) consider ramp duration in the range of hours with a minimum ramp duration of 0.5 hours, which is larger than the maximum ramp duration of the current study.

*Section 3.1:*

*5) Are the 10 min SDs calculated using a sliding 10 min window, or using pre-defined 10 min windows (e.g., 00:00-00:10, 00:10-00:20, and so on)?*

Thank you, good question. The 10-minute standard deviations are calculated from pre-defined windows. This information will be added to the paper.

*6) Some of the selections, e.g., taking 10 min means, using L = 2000m and taking 0.1% of the highest values seem a bit arbitrary. Has some validation been carried out that the selection of high variance events really chooses the important ramp events? Maybe comparison to possible other "ramp event identification" methods?*

Yes some choices seem to be arbitrary and even though we generally try to avoid such choices some are unavoidable. Taking 10-minute averaging times is often argued as a good choice within wind energy, as that time scale is in the 'spectral gap'; it is standard practice in our field. Slower (>10 minute) ramps may affect production, but do not appreciably affect loads. The scale L=2000m is at the lower end of the mesoscale range, and in the current study we want to exclude the largest mesoscale fronts from the characterization. Taking 0.1% is an arbitrary threshold, but it only influences the number of detected events. A comparison with other ramp detection methods might be possible if the existing methods (Gallego et al 2013 and Sevlian and Rajagopal 2013) were adopted to the current time scale of interest, i.e. time scales of 10 minutes or less. In the current study such a comparison is not made, as we choose to focus on a comparison of the ramp characteristics with the IEC ECD characteristics.

*Section 3.1/References:*
*7) Reference "Hannesdóttir et al., 2018": When clicking the DOI link, I get to paper "Extreme fluctuations of wind speed for a coastal/offshore climate: statistics and impact on wind turbine loads"; whereas in the Reference list it reads: "Extreme wind fluctuations: joint statistics, extreme turbulence, and impact on wind turbine loads". Is this the*

*correct reference? (the title of the reference should be fixed anyway)*

Thank you. Yes, this needs to be changed. The reason for this error is that this paper is under review and the title has changed during the review process.

*Figure 4:*
*8) What does the proposed method do if there are two interesting ramp events in the same 1800 s window?*

Good question. Then only the 'biggest' ramp would be characterized (if they are separate). In the wavelet-transform step the maximum wavelet coefficient of each 1800 s window is used to identify the scale and time of the ramp, so if there ever were two ramps, only the maximum wavelet coefficient would be used.

*Figure 7:*
*9) Maybe it would be better to show frequencies instead of counts in Figure 7? As now the counts don't seem very comparable due to different measurement periods.*

Good point. Yes, we will update Figure 7 accordingly.

*Section 3.4/Discussion:*
*10) Are the most interesting ramp events such where both wind speed and direction change quickly? It seems that now only wind speed is used when finding the interesting ramp events (and wind direction change is then simply calculated for these already selected events). But could a combination of wind speed and direction be considered in the ramp event detection method to find the most interesting events considering both wind speed and direction change?*

That is definitely a possibility and an interesting suggestion. Perhaps one could detect the events with a threshold on the direction change and only characterize these

events. That might lead to a smaller sample of detected ramps though. We will look into including this in future work.